# Cross-subsidies are a viable option to fund formal pit latrine emptying services: Evidence from Kigali, Rwanda

Jonathan Wilcox[1]*, Nicholas Kuria[2], Bruce Rutayisire[2], Rachel Sklar[3], Jamie Bartram[1], Barbara Evans[1]

1 School of Civil Engineering, University of Leeds, Leeds, United Kingdom, 2 Pit Vidura, Kigali, Rwanda, 3 Program For Reproductive Health and Environment, University of Caalifornia, San Francisco, CA, United States of America

* cnjdtw@leeds.ac.uk

**Data Availability Statement:** All relevant data are within the manuscript and its Supporting Information files.

## Abstract

Pit latrines are the most common household sanitation system in East African cities. Urbanisation reduces the space available for new latrines to be constructed when pits fill and they increasingly require emptying. But formal services that empty and transport sludge to safe disposal or treatment are often unaffordable to low-income households. Cross-subsidies have been suggested to fund services for low-income households but there are no academic studies assessing this funding mechanism. This study analyses empirical financial and operational data shared by a formal service provider in Kigali, Rwanda who is establishing a cross-subsidy model between corporate and high-income households, and low-income households in informal settlements. A semi-mechanical method is used to serve households which cannot be accessed from the road by an exhauster truck. We find that mechanical emptying is gross profitable when exhauster trucks are fully used, particularly large volume and corporate customers. Transferring sludge between vehicles for efficient transport reduces average cost. Cross-subsidies are found to be a viable funding method and a ten-fold increase in mechanical emptying by the service provider would generate 466,876 Int$ (2022 international dollars) gross profit to fund a cross-subsidy for all low-income households in Kigali which require semi-mechanical emptying. This study highlights the opportunities that city authorities have to organise funding to cross-subsidise emptying for low-income households. In addition, by using data from operational records rather than self-reported estimates the reliability of cost estimates is in improved. Further research is required to understand customer group size, demand and emptying frequencies to determine the structure of a citywide cross-subsidy.

## Introduction

Pit latrines are the most common sanitation system in East and Southern African cities [1]. When pits are full the preferred option is to seal and dig a new pit. But as cities become more

**Funding:** This research was made possible through a grant provided by New Venture Fund (https://newventurefund.org/) to Pit Vidura (NK, BR and RS). This work was also supported by the UKRI Engineering and Physical Science Research Council (EPSRC) (https://www.ukri.org/councils/epsrc/) through a Ph.D. studentship received by the first author (JW) as part of the EPSRC Centre for Doctoral Training in Water and Waste Infrastructure and Services Engineered for Resilience (Water-WISER). EPSRC Grant No.: EP/S022066/1. The funders had no role in study design, data collection and analysis, decision to publish, or preparation of the manuscript.

**Competing interests:** JW is an independent consultant for Pit Vidura. BR, NK and RS established Pit Vidura's service which is analysed in this paper. The remaining authors declare no conflict of interest (financial or other) that might be construed to have influenced the content of this paper.

densely populated there is less space available for new pits and they therefore require emptying [2]. The private sector generally provides mechanical emptying services using exhauster trucks [3]. Formal service providers are licensed by the city authority which requires them to transport sludge to safe disposal or treatment [4]. But many pit latrines cannot be emptied by exhauster trucks because they are inaccessible from the road network or the sludge is too thick to pump [5]. Instead, households empty their own pits or rely on informal manual emptiers who empty and dispose of sludge in ways that threaten public and environmental health [6]. There are few formal service providers which offer alternatives to mechanical emptying and those that are available are often unaffordable. Additional funding or a subsidy is required to lower the price and increase the number of households able to access formal services [4, 7–9]. Subsidies are funds which are directed from government to service providers and customers, or between customer groups (cross-subsidies) to fill the gap between service providers' costs and the user payment. They can be explicit (through funding transfer) or implicit (for example where inputs such as energy are under underpriced). Subsidies can also be internal (funded from within the service providers' own business) or external (from government to service providers) [10]. This study considers the potential for internal cross-subsidies to lower tariffs for low-income households, thereby replacing informal emptying and pit sealing using gross profits from services provided to other customers.

Designing subsidies requires data about the service cost, target services and users, and affordability [10]. The funding gap for low-income households has been assessed using voucher schemes in Malawi [7], Rwanda [8] and Kenya [4]. Public funding is a common funding source for subsidies and in Wai, India the city authority have implemented a scheduled emptying programme funded by a progressive property tax where households do not directly pay for services [11]. Cross-subsidies have also been suggested by using revenue from profitable customer groups such as institutions [12]. The potential for cross-subsidies for low-income households has also been assessed from water-supply customers in Kenya [13] and from high-income customers in Uganda [14]. In Rwanda the national regulator is planning to introduce a scheduled emptying programme for all households funded by a sanitation fee on the water tariff [15]. In Bangladesh the SWEEP project has implemented an implicit internal cross-subsidy where the city authority leases exhauster trucks on the contractual condition that 30% of customers are low income and are charged a lower volumetric tariff [16]. But no peer-reviewed studies have described or assessed implementing a cross-subsidy specifically for pit emptying services, and few of the current studies are based on empirical evidence from financial statements [17].

The aim of this study is to assess the viability of cross-subsidies to fund formal pit latrine emptying and transport services. This is achieved by analysing operational and financial records shared by Pit Vidura, a social enterprise in Kigali, Rwanda who is implementing an explicit internal cross-subsidy between different customer groups [18]. The study describes the revenue streams from different customer groups, calculates the direct and indirect costs of different services, and estimates the operating scale required to generate the gross profit to cross-subsidise semi-mechanical emptying services and to increase coverage amongst low-income households to replace informal manual emptying and pit sealing. The aim of this study is to assess the viability of cross-subsidies to fund formal pit latrine emptying and transport services.

## Methods and materials

### Study context

Kigali has no centralised sewer system and is rapidly urbanising [19]. Regulation in the city requires that all pit latrines must be mechanically emptied [20]. Formal businesses offer

mechanical emptying services using exhauster trucks and most sludge is transported 20 km from the city centre to the city dumpsite [2]. Pit latrines are very common amongst low-income households living in informal settlements which are characterised by challenging conditions: steep slopes, flood plains, swamps and rocky grounds [19]. This makes emptying sludge from pit latrines challenging.

Pit Vidura is a social enterprise and was founded in 2016 to improve public and environmental health by providing emptying services to low-income urban households [18]. They are the only formal business in Kigali serving households which cannot be accessed by exhauster trucks. Their business model is to use operational data and research to reduce costs, and to establish an internal cross-subsidy between corporate and high-income household customers, and low-income households. Grants have funded operations, and ongoing research and development. Pit Vidura identify as a social enterprise because they provide formal services to households which are not served by the for-profit service providers in Kigali.

Pit Vidura has three exhauster trucks of varying volumes which offer services to different customer groups [21, 22]. The largest is primarily intended to serve corporate customers and the middle-sized truck primarily households. The smallest serves households which are inaccessible to the other two. Pit Vidura is the first company to own a small exhauster truck in Kigali and this has allowed them offer mechanical emptying to households that would otherwise use informal manual emptiers. Where an exhauster truck cannot directly access a facility to pump the sludge, facilities are emptied semi-mechanically using a barrel-based method: a portable vacuum pump empties the sludge into barrels that are carried to a nearby location where sludge is transferred to the small exhauster truck. Before purchasing the small exhauster a rented flatbed truck was used to transport sludge in barrels to the dumpsite.

The large exhauster truck is the most fuel-efficient—in terms of fuel consumption per sludge volume per distance. It is used to transport sludge collected by the small exhauster truck to the dumpsite. Sludge is also occasionally transferred to the medium exhauster truck for temporary storage before being transferred to the largest exhauster truck for transport to the dumpsite.

Emptying requests are coordinated by a call centre which works with a pit evaluator to identify the most suitable emptying method for the customer based on exhauster truck availability [22]. Although each exhauster truck has a notional customer group they each serve corporate and household customers, and both sealed tanks (called septic tanks in Kigali) and pit latrines as required. Rental vehicles are used to fulfil requests when no suitable exhauster truck is available. The tariff is based on customer type (corporate or household), volume, emptying method (mechanical or semi-mechanical), and distance to the dumpsite. Lower volume emptying has a higher volumetric tariff and corporate customers pay a 10% premium. 18% VAT (value-added tax) is paid on all emptying jobs [23].

## Data familiarisation

This study utilises data collected and shared by Pit Vidura: company profit and loss statements; asset depreciation records; and downloads from two Customer Relationship Management (CRM) systems detailing operational records. Data were treated as secondary because they were not produced specifically for this research. Permission was provided by Pit Vidura management for the data to be used in this study.

Profit and loss statements are available from 2018. Statements include all direct (S1a Table in S1 File and S1a Dataset in S1 Data) and indirect costs (S1b Table in S1 File and S1a Dataset in S1 Data), and revenue (S1c Table in S1 File) for the financial year. Most direct costs are attributed to a specific exhauster truck. Pit Vidura account for exhauster trucks, pumps

and major repairs based on straight line depreciation and a four-year useful life. Direct staff costs are emptiers and drivers, where the emptying team leader and driver are paid a salary, and any additional emptiers (notably for semi-mechanical emptying) are paid a daily wage. Indirect staff costs are a general manager, research engineer, call-centre agent and an accountant/planner, who are all paid a salary.

Pit Vidura coordinate emptying requests using two CRM systems: a spreadsheet log and a cloud-based software (Salesforce), with records from 2016 and 2019 respectively (S1c and S1d Dataset in S1 Data). The systems are both managed by a call-centre agent and an accountant/planner. The spreadsheet log records: request identification number, customer identification number, customer type (household or corporate), customer status (first-time or repeat), containment type (pit latrine, soakaway or septic tank), exhauster truck, emptying method (exhauster truck or portable vacuum pump and barrels), emptying date, customer location, number of trips, price, and the number of barrels emptied if using the portable vacuum pump. The cloud-based software records the same and additionally the number of staff (drivers and emptiers) assigned to each job. Both CRM systems collect other data fields that were not used in this study. Both CRM systems record emptying requests which are not converted to completed jobs. Costs associated with emptying requests that are not converted are included as indirect costs. Only completed jobs where sludge emptying occurs are included when calculating the average cost per job. Personal identifiers were removed from both datasets.

## Secondary analysis

**Organising secondary data.** Details of emptying jobs from the two CRM systems were organised by year, customer type, exhauster truck, emptying method, and customer status. Total annual revenue and average revenue per job were calculated (S1a Table in S1 File). Jobs recorded without a price, customer type or emptying date were assumed to be unconverted, and jobs with zero revenue were assumed to be follow-ups from previous jobs. Large and medium exhauster truck jobs were grouped together as high-volume emptying, and all small exhauster truck jobs were grouped as low-volume emptying.

Costs were thematically analysed and grouped into the eight largest direct and eight largest indirect cost categories. Direct costs are those that can be attributed to a specific service, for example exhauster trucks, fuel and wages (S1a Table in S1 File); indirect costs are those that are shared amongst services, for example office rent, marketing and management salaries (S1b Table in S1 File). Average cost per job for each year was calculated pro-rata based on the total number of trips completed for each year, exhauster truck and emptying method, and customer type (S1d Table in S1 File). Costs for mechanical and semi-mechanical emptying by the small exhauster truck are combined in the data and were split pro-rata assuming that four mechanical and two semi-mechanical jobs can be completed per day. Emptiers' wages were shared based on the total number of emptier working days where two and five emptiers are required for mechanical and semi-mechanical jobs respectively. For 2018 and 2019 data for trips per job are not available so the average from 2020 to 2022 is used. Direct costs that are not assigned to an exhauster truck were split pro-rata based on the number of trips made by each exhauster truck.

**Modelling assumptions.** To estimate the operating scale and cross-subsidy required to replace informal emptying and pit sealing, values are used from literature and assumptions made to model cost efficiencies from operating at a large scale. These are based on assumptions made by others estimating the total funding gap for sanitation services [4, 8, 24].

Mechanical emptying jobs were assigned a sludge volume based on exhauster truck capacity ($20 \text{ m}^3$, $10 \text{ m}^3$ and $5 \text{ m}^3$) and $2 \text{ m}^3$ for semi-mechanical emptying jobs (S1e Table in S1 File).

The cost of the large exhauster truck transporting sludge for the small exhauster truck was modelled based on the additional trips required and relative capacities, for example 2 m$^3$ transferred to 20 m$^3$ is equivalent to 0.1 additional trips. Sludge transfer cost was discounted pro-rata from all direct cost categories based on the additional number of trips.

Economies of scale are modelled based on higher rates of exhauster truck and emptier use [8]. Full daily use was assumed to be four jobs per day for mechanical emptying (including trips for transferring sludge by the large exhauster truck) and two trips per day for semi-mechanical emptying as the overall emptying process has a longer duration [21]. Full daily use assumptions were verified by operating records and confirmed by literature [3]. Exhauster trucks were assumed to operate 250 emptying days per year to allow for maintenance downtime. The quantity of exhauster truck type was increased equally, for example four small, four medium and four large. The corporate/household split was assumed to be the same as the year in which that vehicle completed the most jobs. The costs for each method were modelled based on the year in which the most jobs were completed (2021 for the medium exhauster truck and 2022 for all others), in order to provide the most data to produce a more reliable cost estimate. Direct cost categories were assumed to be either fixed (repair and maintenance, depreciation, drivers' salary, and other) or variable and proportional to the number of trips. For indirect costs it was assumed an additional call-centre agent would be required to co-ordinate emptying requests between 1500 and 3000 jobs per year. Above 3000 jobs per year it was assumed that two additional agents (call-centre and accountant) would be required for each additional 3000 jobs per year. All other indirect costs were assumed to be fixed.

Data about market size, structure, and willingness to pay in Kigali was taken from literature. A revealed and stated preference study conducted by Burt, Sklar and Murray [8, p. 9–10] found that 87% of households seal pits or use informal emptiers, and that a 63% tariff reduction for semi-mechanical emptying would be required for low-income households to stop sealing pits and instead use formal semi-mechanical emptying services (S1e Table in S1 File). The same study also found that the mean emptying frequency for household pit latrines is 8.7 years, and that the mean low-income household size is 6.1 people. We also assume that: four low-income households share a pit latrine [19]; Kigali has a total population of 1,745,555 and 47% of households use shared pit latrines with constructed floor slabs which can be emptied semi-mechanically [25]; and that pit latrines (shared or private) without constructed floor slabs cannot be emptied using formal methods, and that all others can be mechanically emptied [5].

All financial values from secondary data and literature were converted from Rwandan Francs (RWF) to international dollars (Int$) based on consumer price index (CPI) and purchasing power parity (PPP) in 2022 [26]. Values from 2023 were converted assuming average CPI and PPP from the preceding three years.

## Results

### Establishing different revenue streams

Fig 1 shows that revenue from household first-time customers has increased steadily since 2016. Repeat household customers are a notable portion of total revenue (27% in 2023). Revenue from corporate customers has increased steadily since the purchase of the large exhauster truck in 2019 and in 2023 repeat corporate revenue accounted for 45% of total revenue (S1c Table in S1 File).

In the first 18 months of all three exhauster trucks being available (from November 2021) the large exhauster truck completed almost as many household (n = 322) as corporate (n = 371) jobs (S1f Table in S1 File). The medium exhauster truck completed more household

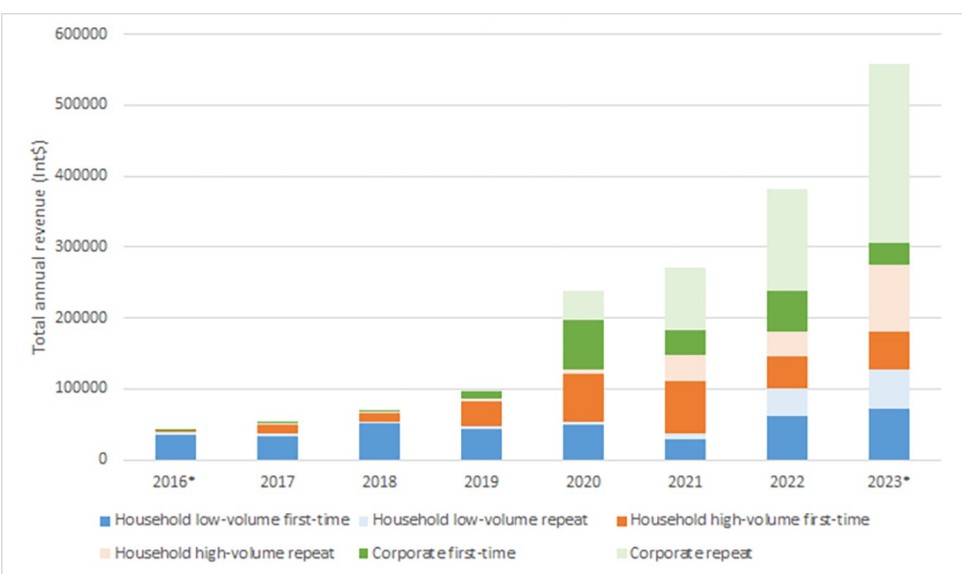

**Fig 1. Pit Vidura total annual revenue 2016 through 2023 organised by customer type, customer group and customer status.** Type (household or corporate), group (high–volume or low–volume) and status (first–time or repeat). 2016* and 2023* are partial years based on 2 and 6 months respectively. Values are 2022 international dollars (Int$).

(n = 262) than corporate (n = 79) jobs. The small exhauster truck completed a small number of corporate (n = 27) but mostly household mechanical (n = 625) or household semi-mechanical (n = 132) jobs. A flatbed truck was rented for a small number of jobs (n = 15) when the small exhauster truck was not available and an exhauster truck was rented to fulfil large volume emptying requests when the large and medium exhauster truck were not available (n = 10). All exhauster trucks operated at fully daily use on some days but have capacity to increase the number of jobs.

## Costs of formal services

Table 1 shows the average cost per job broken down by direct and indirect costs in the year that each truck completed the most jobs: 2021 for the medium exhauster truck and 2022 for all others. The large exhauster truck completed 442 jobs in 2022—its busiest year. Transport related costs (fuel, repair and maintenance, depreciation, drivers' salaries and rental) accounted for 87% of directs costs.

During 2022 the medium exhauster truck completed fewer jobs (n = 157) than in previous years (n = 395 in 2021) because it was unavailable for six months whilst undergoing major repairs. The medium exhauster truck has also been used as an intermediate storage tank for sludge collected by the small exhauster truck for transport to the dumpsite by the large exhauster truck. Transport related costs accounted for 85% of direct costs in 2022—similar to the large exhauster truck.

The small exhauster truck completed 508 jobs in 2022, mostly mechanical (n = 384) rather than semi-mechanical (n = 89) emptying for household customers. For mechanical emptying, vehicle depreciation accounted for a large proportion (39%) of direct costs. Driver costs are comparable to the large and medium exhauster trucks' but emptiers' wages are lower despite more emptiers being used. This is because high-volume jobs often require night work or overtime to complete jobs, which requires multiple trips to the dumpsite which is open 24 hours per day.

**Table 1. Average cost per job for emptying of pits and tanks by Pit Vidura, broken down by direct and indirect in 2022.** All costs are 2022 international dollars. Customer and emptying methods with fewer than 50 jobs in 2022 not shown. *values for the medium exhauster are from 2021 because in 2022 it was unavailable for six months undergoing major repairs.

| | Direct costs | | | | | | Indirect costs | |
| --- | --- | --- | --- | --- | --- | --- | --- | --- |
| Customer group | Corporate | | Household | | | | | All |
| Emptying method | Large exhauster truck | Medium exhauster truck* | Large exhauster truck | Medium exhauster truck* | Small exhauster truck | Semi-mechanical emptying | | All |
| Jobs per year | n = 267 | n = 35 | n = 175 | n = 111 | n = 384 | n = 89 | | n = 1130 |
| Fuel | 161 | 52 | 126 | 52 | 21 | 33 | Staff salaries | 72 |
| Repair and maintenance | 96 | 75 | 76 | 75 | 16 | 26 | Marketing and advertising | 15 |
| Vehicle depreciation | 33 | 40 | 26 | 40 | 52 | 83 | Staff expenses | 9 |
| Vehicle rental | 33 | 0.0 | 26 | 0.0 | 2 | 3 | Accounting, consulting and banking | 10 |
| Emptiers' wages | 23 | 24 | 18 | 24 | 8 | 31 | Office | 25 |
| Dumping fees | 18 | 5 | 14 | 5 | 0.9 | 1 | Tax | 40 |
| Drivers' salaries | 18 | 17 | 14 | 17 | 16 | 26 | Communications | 10 |
| Consumables | 5 | 6 | 4 | 6 | 5 | 8 | Other | 11 |
| Equipment | 4 | 4 | 3 | 4 | 7 | 39 | | |
| Other | 2 | 8 | 2 | 8 | 4 | 6 | | |
| Sludge transfer | | | | | 75 | 30 | | |
| *Average direct cost per job* | *392* | *231* | *308* | *232* | *207* | *287* | *Average indirect cost per job* | 192 |

Semi-mechanical emptying has a higher average cost per job than mechanical emptying with the small exhauster truck because fewer jobs can be completed in one day, and semi-mechanical emptying requires more emptiers. Equipment costs for semi-mechanical methods are proportionally higher (18%) than for mechanical emptying methods (all 5% or less) because the additional cost of the portable vacuum pump is included.

Indirect costs account for 37% of total costs in 2022, where the three largest cost categories are staff salaries (19%), tax (10%) and office (6%) (S1b Table in S1 File). In 2019 and 2020 there was a donor funded project for consulting and marketing with large associated expenditure but these are two small categories in 2022 and account for 4% of total costs. Since 2019 both the total indirect costs and the average indirect cost per job have decreased.

## Implementing the cross-subsidy services for low-income households

Fig 2 shows the financial flows in 2022 between the three customer groups and the five emptying methods. Together all methods generate a 7% gross profit (27,285 Int$). Emptying by the large exhauster truck is the only individual method to generate a gross profit and this is used to cross-subsidise the emptying services provided by the other four methods. Corporate emptying by the large exhauster truck accounts for 37% of total revenue (S1d Table in S1 File). Since 2018 gross profit has increased and indirect costs have decreased. 2022 is the first year that grant funding is only used to fund indirect costs (S1g Table in S1 File).

The financial cross-subsidy provided to semi-mechanical emptying is negligible in 2022 because the service is priced to recover direct costs and the tariff has not been reduced to increase demand.

The large exhauster truck also transported sludge emptied by the small exhauster truck to the dumpsite. This increases the total costs attributed to the large exhauster truck in the

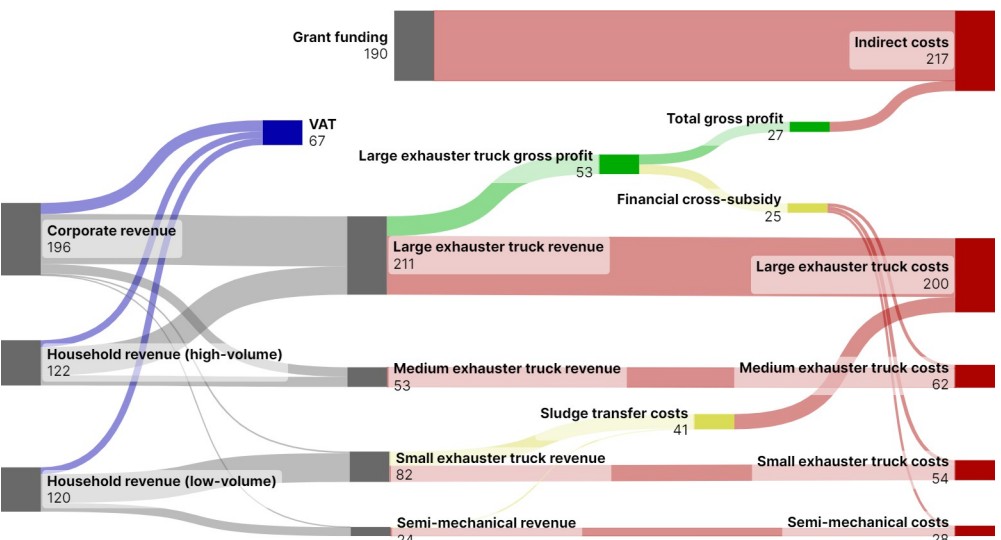

**Fig 2. Pit Vidura financial flows 2022.** Values are 1000 international dollars 2022. Revenue is shown in grey, Value Added Tax (VAT) in blue, gross profits in green, costs in red, and cross–subsidies in yellow.

financial records but they are not related to the high-volume emptying service that it provides. The total overall cost would be higher if the small exhauster truck transported sludge to the dumpsite because it is less fuel efficient at sludge haulage. Sludge transfer accounts for 34% of the small exhauster truck's direct costs (Table 1).

In 2022, VAT collected from all services was equivalent to 20% of total direct costs and is greater than the financial cross-subsidy provided for semi-mechanical emptying services (S1g Table in S1 File).

### Estimating the cross-subsidy required to replace informal emptying and sealing

Fig 3 shows the average direct cost and revenue per job for each customer group and emptying method assuming that each vehicle operates at full use and that the tariff for semi-mechanical emptying is lowered to the level require to replace informal emptying and pit sealing. At full use each mechanical emptying method would generate a gross profit and this could be used to fund indirect costs and cross-subsidise semi-mechanical emptying. The large exhauster truck has the largest gross profit and highest gross margin of all customer groups, and could make a proportionally larger contribution to funding indirect costs (S1h Table in S1 File).

If all three exhauster trucks were fully used then transport related costs would account for 68% of direct costs. Costs related specifically to sanitation (for example emptying labour, dumping fees, consumables and equipment) are minor (19%). Therefore from a cost perspective when operating at scale, emptying and transport services are primarily a logistics and fleet management industry, and sanitation is secondary.

Semi-mechanical emptying would require a total subsidy of 466,867 Int$ per year to replace informal manual emptying and pit sealing in Kigali (S1i Table in S1 File). Fig 4 shows the estimated financial flows between customer groups if Pit Vidura operated at a scale to generate enough gross profit for this cross-subsidy, and to fund indirect costs without grant funding. This would require a twelve-fold overall increase in total emptying jobs from 2022 (S1h Table in S1 File). It would also require 19 fully used exhauster trucks in total including 7 small exhauster trucks dedicated to semi-mechanical emptying for low-income households, for 51%

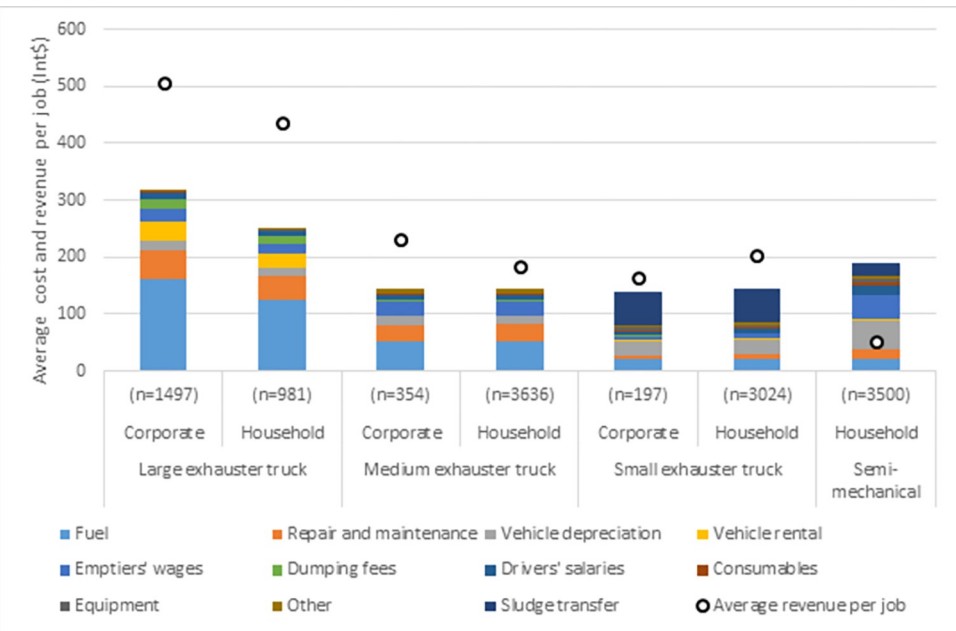

**Fig 3. Average cost and revenue per job for each customer group and emptying method at full vehicle use.** Cost assumes that each vehicle operates at full use. The tariff for semi–mechanical emptying is lowered to the level require to replace informal manual emptying and pit sealing. n = number of jobs per year.

of jobs to be high-volume, and 27% of jobs to be semi-mechanical. Estimated VAT collected from all emptying services (496,637 Int$) at this scale is a similar amount to the financial cross-subsidy required to replace pit-sealing (S1h Table in S1 File).

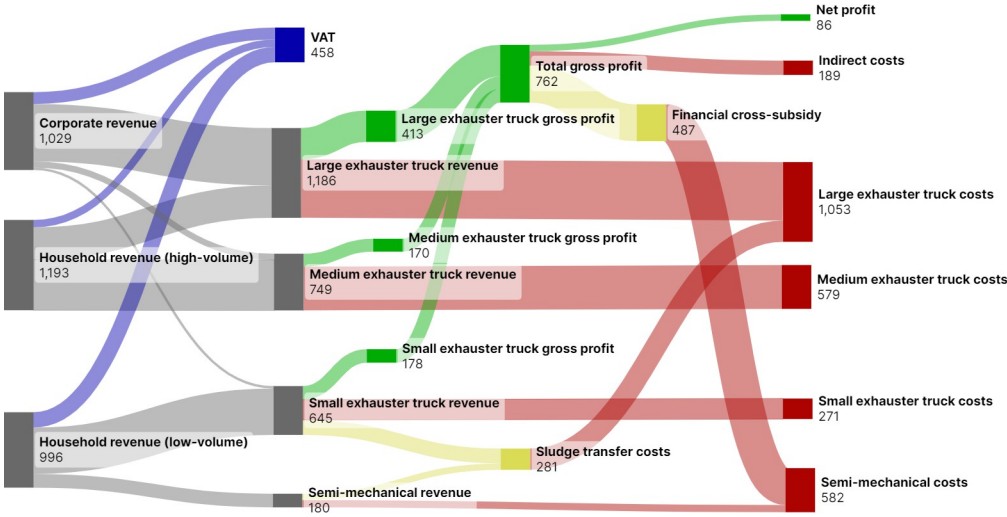

**Fig 4. Modelled financial flows to cross–subsidise semi–mechanical emptying to replace informal manual emptying and pit–sealing, based on Pit Vidura operations in Kigali, Rwanda.** Scale required is 19 exhauster trucks operating at full use (4 large exhauster trucks, 4 medium exhauster trucks, 4 small exhauster trucks dedicated to mechanical emptying, and 7 small exhauster trucks dedicated to semi–mechanical emptying). Values are 1000 international dollars 2022. Revenue is shown in grey, Value Added Tax (VAT) in blue, gross profits in green, and cross–subsidies in yellow.

## Discussion

### Viability of a cross-subsidy to fund semi-mechanical emptying to replace informal manual emptying

Pit Vidura have developed services for specific customer groups by combining different vehicles and emptying methods to serve different customer groups [22]. In 2022 Pit Vidura generated an overall gross profit for the first time but this was all derived from the largest exhauster truck. Semi-mechanical emptying services are offered to low-income households at close to direct cost price. But at this price demand for formal services is low and informal emptying is preferred by most households [8]. A ten-fold increase in mechanical emptying jobs is required to generate sufficient gross profit to fund the cross-subsidy for semi-mechanical emptying for low-income households to replace informal emptying and pit-sealing, and to fund indirect costs without reliance on grant funding.

Despite the higher volumetric tariff for semi-mechanical emptying compared to mechanical emptying, semi-mechanical services operate at a loss. This is partly due to limited economies of scale but also because semi-mechanical emptying has a much higher volumetric cost than mechanical emptying as it is more laborious and fewer jobs can be completed in a working day [22]. As a social enterprise, Pit Vidura voluntarily implement their cross-subsidy. This differs from other service providers in Kigali (and more widely) which are profit driven. Tariffs for high volume mechanical emptying are determined through competition [27]. Other service providers also have lower costs by offering a less professional service, not operating a call centre, and are able to reinvest net profits. This creates a challenging environment to implement a cross-subsidy model and Pit Vidura must be more cost efficient than their competitors.

Pit Vidura's tariff structure contrasts with the SWEEP project in Bangladesh where low-income customers have a 40% lower volumetric tariff than high-income customers [16]. In comparison the tariff reduction required to replace pit sealing in Kigali would establish a similar volumetric tariff to the large exhauster truck, and would be about 50% lower than the volumetric tariff for mechanical emptying by the small exhauster truck. Our estimates suggest that this could be replicated by Pit Vidura but that it would require a large increase in operating scale to achieve the efficiencies required to minimise operating costs and also to be financially viable without depending on external funding. In the SWEEP project exhauster trucks are leased to service providers on the contractual condition that 30% of customers are low-income and charged the lower volumetric tariff. This is lower than the proportion of low-income customers that can be funded using a cross-subsidy as indicated by our estimate, especially when taking into consideration that in the SWEEP project all emptying is mechanical which is lower cost, that exhauster trucks are leased to service providers to minimise financial risks, and that exhauster trucks are lower cost in Asia than Africa [28]. The SWEEP project also demonstrates an alternative service model (an implicit internal subsidy) to Pit Vidura's where the city authority implements a cross-subsidy by regulating exhauster truck rental price and the associated contractual conditions, as opposed to the service provider implementing an explicit internal cross-subsidy.

Informal manual emptying is a close substitute service for semi-mechanical emptying in informal settlements and is common in Kigali. Informal emptiers do not pay VAT or any other taxes because regulation prohibits their methods and they cannot be licensed [2]. Informal emptiers also do not have the considerable costs related to transporting sludge to disposal or treatment. Together this contributes to keeping informal emptying below the market price for formal emptying, and as a close substitute lowers demand for formal services [8, 27]. At the same time households have also expressed a preference for services that treat sludge and protect workers [8]. Demand for formal services may increase as they become more common due

to the increasing returns to scale for households and the perceived benefits [29]. However other factors beyond price influence the use of formal services and lowering the tariff may not be sufficient to replace informal emptying or pit sealing [7]. Regulators could consider creating more favourable tax conditions for formal service providers to support them to compete with informal emptiers (for example reducing import taxes on exhauster trucks, insurance discounts or vehicle licenses), as in other countries [3].

Formal emptying services in Kigali are nascent [2]: there are relatively few formal service providers and as a feature of recent and rapid urbanisation many pits have not yet been emptied. Only one in three pits have been reported as previously emptied, possibly because many households having high volume pits [8]. Ross and Pinfold [27] estimated 1,300 mechanical emptying jobs per year were undertaken in Kigali by 10 partially used exhauster trucks in 2017. This contrasts with our estimate to fully cross-subsidise semi-mechanical emptying: 10,500 mechanical jobs per year by 12 fully used exhauster trucks (equivalent to 875 per truck per year). The census data indicate that about half of households use either private pit latrines or flush systems (septic tanks) [25, Table 70] and these can probably be emptied mechanically [5, 27]. Together this indicates that there may be enough households and institutions requiring mechanical emptying in Kigali to fund the required cross-subsidy, particularly because systems that can be emptied mechanically generally have a higher emptying frequency [30]. Our findings are consistent with previous studies which found emptying to be profitable when operating at scale and serving high volume or institutional customers [3, 16, 17, 28]. Other cities should also recognise that the opportunity to do this varies by location depending on demand and market share of different customer groups and the cost of delivering services, and that this may require additional funding, particularly if there are not many profitable corporate or high volume household customers to serve.

Previous studies have highlighted that often household systems cannot be emptied by formal service providers because latrine structures are unstable especially during the rainy seasons, there is too much trash or the sludge is too thick [5, 19]. In Kigali accepting improved manual methods, that are suitable for more systems than the semi-mechanical methods required by regulation [20], is likely to be required to further extend formal emptying services [22]. In addition, improving household systems so that they are easier to empty increases the likelihood that formal services are used [31]. In nearby Kampala, Uganda the city authority has supported landlords to upgrade latrines [32] recognising the challenges facing tenants to improve household systems [19]. In Kenya it has been proposed for formal service providers to work with informal emptiers to transport sludge to safe disposal or treatment [33] but this may not be politically acceptable in Rwanda or elsewhere. The proportion of institutional customers, and household sanitation system types and condition will determine the viability of an internal cross-subsidy model but this is outside the influence of the service providers. To extend formal services to households that cannot be emptied semi-mechanically will require a combination of containment system improvements and also working with manual emptiers to improve and formalise their service.

## Funding services through taxation

Clean water supply and environmental treatment for non-profit making purposes are zero-rated in Rwanda but not faecal sludge emptying (RRA, 2012). In 2022 VAT payments by Pit Vidura were equivalent to 19% of total direct costs. Our estimates indicate that VAT payments are similar to the cross-subsidy required to replace informal emptying and pit sealing. This highlights that VAT revenue could either be ringfenced and used as an alternative to a cross-subsidy to fund semi-mechanical emptying, or that emptying could be VAT zero-rated and an

alternative sanitation tax be introduced for some customer groups to cross-subsidise semi-mechanical emptying. Kigali has a solid-waste collection service funded through a monthly fee [2] and the regulator is planning to introduce a sanitation fee to the water tariff [15].

In Wai and Sinnar, India a progressive household sanitation tax has been used to fund a privately contracted, scheduled emptying service. The cities are split into three zones and one zone is emptied each year. 6800 households were emptied in Wai over three years with a 95% acceptance rate [11]. Scheduled emptying gives service providers long-term visibility of emptying jobs which allows them to optimise and minimise costs [11, 16]. It also removes the urgency for most households when pits are full, which reduces the possibility of corruption from emptying teams [2]. This model contrasts with both the SWEEP project and Pit Vidura because the government coordinate the implicit cross-subsidy between households by having a higher tax rate for larger properties and also because the households do not directly pay the contracted service provider [11].

## Effective fleet management is required to minimise costs

Costs associated with transport (capital investment, maintenance, fuel and drivers) account for a large proportion of total costs when operating at a large scale and trucks are fully used. Fuel alone is 30% of total costs which is consistent with previous studies [3]. But the costs related specifically to sanitation (emptying pumps, emptiers' wages and PPE) are relatively small (14%). This highlights the importance of thinking of faecal sludge emptying and transport services as a haulage industry and of effective fleet management in minimising costs and extending services.

Exhauster trucks are generally second hand and imported in Sub-Saharan Africa, and in poor mechanical condition [3]. Both operating and capital costs are higher than in South Asia [17, 28]. This is consistent with Pit Vidura's experience where second hand trucks have relatively high ongoing maintenance costs. This is the basis for the use of the straight-line depreciation of the truck value over four years in the Pit Vidura accounts, a reasonable assumption for second-hand vehicles in poor condition as they are unlikely to have the same lifetime as a new vehicle. Regulating vehicle condition and facilitating access to credit to purchase trucks in better condition could enable service providers to lower long term average costs [3].

Using grant funding Pit Vidura has purchased all three vehicles directly. This contrasts with the model used by the SWEEP project where the city authority purchased the trucks (with philanthropic and NGO support) and leases them to service providers [16]. For SWEEP this was part of the project design to remove the financial risk of owning vehicles from the service provider. This is similar to Pit Vidura who have avoided the considerable costs and risks associated with financing truck purchase using loans.

Our estimate assumes that the exhauster trucks are fully used, to minimise operating costs and the required cross-subsidy [8]. Maximising truck use and minimising operating costs was one of the arguments for adopting a scheduled desludging approach in Wai and Sinnar [11]. This gives service providers' long-term visibility of emptying requests so that they can fully use trucks and effectively manage maintenance. This contrasts with the situation in Kigali, where Pit Vidura and other service providers do not fully use exhauster trucks [27]. Regulators could enable this efficiency by limiting the number of formal service providers to increase their market share. Development actors have begun to assist service providers to access bank loans to finance truck purchase [34], following calls for this support [3].

Transfer stations have been widely proposed as a method to minimise emptying and transport costs [3, 28, 35] and as an option specifically for Kigali [2, 27]. But they have been unsuccessful elsewhere, largely because of objections from the local community [36]. Pit Vidura

have achieved the same outcome by using the most fuel efficient vehicle to haul sludge to the dumpsite but also by using exhauster trucks as mobile transfer stations by transferring sludge between trucks for intermediate storage. This innovation avoids the need for permanent or even semi-permanent tanks. Mobile transfer stations and haulage function in a similar way to an implicit, internal cross-subsidy where inputs are underpriced: the cost of transport for semi-mechanical is lower and if informal manual emptiers were to transport sludge to the dumpsite they would incur a higher cost.

## Grant funding has enabled Pit Vidura to establish the cross-subsidy

Grant funding has enabled Pit Vidura to develop a business model which addresses the access, availability and affordability challenges of providing formal services to low-income households [6]. It allowed them to do several things that other service providers are unable to fund: identifying a suitable emptying method for low-income households which is compliant with local regulations [22]; compete with the other formal service providers on price who are offering a less professional service [27]; initially avoiding loans but building capacity and creditworthiness towards being able to use loans to purchase exhauster trucks in the future [28, 34]; aspects of the business model that are required for a large scale service including the call centre, professionalising services (for example using a call centre to manage and interact with customers); having a research and innovation approach [12]; and to disseminate learnings [21, 22].

It is important to recognise the role that grant funding has played because it is not universally or widely replicable. Instead, consideration is required as to how public funding can be similarly used.

## Introducing a regulated cross-subsidy in Kigali

In Rwanda the national regulator is planning to introduce a sanitation fee onto water bills from both mains and standpipe customers to fund citywide emptying services. The national utility will be accountable for ensuring services are delivered, with the option to subcontract them to private service providers. The national government has committed to funding treatment, with long held plans to construct a treatment plant in the city [15].

Non-user valuation of the indirect benefit from improving sanitation in informal settlements is a large contributor to the overall benefit. Survey responses from Kampala indicated that non-user valuation would be able to cover the majority of service costs [14]. Similar research from Kenya indicated that high-income water utility customers are willing to pay for a cross-subsidy for low-income customer [13]. This suggests that it may be possible for the Rwandan regulator to introduce a pro-poor cross-subsidy for emptying services through the sanitation fee on the water tariff.

The proposal from the regulator takes a similar format to the service model in Wai and Sinnar [11], except the proposed cross-subsidy is explicit and internal to the utility through the water tariff, rather than explicit and external through public funding and housing tax. Irrespective of the contracting model preferred by the utility, effective regulation will be required to manage the tension between informal and formal service providers to ensure services are accessible and equitable.

## Limitations

The limitations of this study should be acknowledged when interpreting the findings. The study uses data from a single city and service provider—consideration should be made for contextual differences when generalising results to other cities, for example different costs and willingness to pay. Cost analysis is based on financial reports rather than whole-life costing

which may provide more accuracy through a long-term perspective but would rely on more assumptions, for example about vehicle lifespan and maintenance. Analysis is based on the average cost for each customer group, rather than by establishing the cost for each individual emptying job, which prevents analysis within customer groups. Small sample sizes prevented analysis of the relationship between vehicle condition, and maintenance and operating costs which likely overestimates and underestimates the costs of newer (large and small exhauster truck) and older vehicles (medium exhauster truck) respectively because the same depreciation lifespan is used irrespective of vehicle condition. The customer sample is not representative of the city but biased towards those with a preference for a professional service provider. The estimate for eliminating informal emptying and pit-sealing is based on assumptions derived from census data. This probably underestimates the requirement for semi-mechanical emptying because households are likely to over report construction of septic tanks as they are required by local regulation. No data is available for institutional systems, emptying frequencies and willingness to pay. Potential cost savings from transport path optimisation have not been included in analysis [37, 38].

## Conclusion

This study presents the first empirical analysis of a formal pit latrine emptying and transport service provider establishing an explicit internal cross-subsidy to low-income households. Findings are consistent with previous research which found that mechanical emptying and transport services are gross profitable when provided efficiently and at scale, particularly when including institutional customers. Profits can be used to cross-subsidise semi-mechanical emptying for low-income households to increase coverage of formal services. We find that replacing informal emptying and pit sealing in Kigali would require a ten-fold increase in mechanical emptying jobs completed by Pit Vidura to cross-subsidise semi-mechanical emptying.

The private sector is not incentivised to provide formal services in informal settlements and regulation is required to manage the tension between affordability, service quality and inclusivity. Regulators should organise funding flows between customer groups and lower tariffs for semi-mechanical emptying for low-income households to replace informal emptying. Regulators should also enable service providers to minimise costs by fully using exhauster trucks and efficiently coordinating services, reducing the need for subsidies. Alternative options for households that cannot be emptied by formal methods must be developed to fully replace informal emptying and pit sealing. Further research is required to quantify customer groups and distribution, sludge accumulation rates and emptying frequencies, and institutional and high-income household willingness to pay to determine how to structure citywide funding. Depending on the market structure it may be possible replace informal emptying and pit sealing without external funding.

Cross-subsidies are a viable option to fund pit latrine emptying services in informal settlements and cities should consider it in addition to other activities to achieve universal coverage of safely-managed urban sanitation.

## Supporting information

**S1 Checklist. Inclusivity-in-global-research-questionnaire.**
(DOCX)

**S1 File.**
(DOCX)

**S1 Data.**
(XLSX)

## Acknowledgments

Thank you to Leonie Hyde-Smith, Virginia Roaf, Carlo Prato and Elizabeth Tilley for providing comments on a manuscript draft.

## Author Contributions

**Conceptualization:** Jonathan Wilcox, Nicholas Kuria, Bruce Rutayisire, Rachel Sklar, Jamie Bartram, Barbara Evans.

**Data curation:** Jonathan Wilcox, Nicholas Kuria, Bruce Rutayisire, Rachel Sklar.

**Funding acquisition:** Nicholas Kuria, Bruce Rutayisire, Rachel Sklar, Barbara Evans.

**Investigation:** Jonathan Wilcox, Nicholas Kuria, Bruce Rutayisire.

**Methodology:** Jonathan Wilcox.

**Project administration:** Jonathan Wilcox, Nicholas Kuria.

**Resources:** Nicholas Kuria, Bruce Rutayisire.

**Supervision:** Rachel Sklar, Jamie Bartram, Barbara Evans.

**Validation:** Nicholas Kuria, Bruce Rutayisire, Rachel Sklar.

**Visualization:** Jonathan Wilcox.

**Writing – original draft:** Jonathan Wilcox.

**Writing – review & editing:** Jonathan Wilcox, Nicholas Kuria, Bruce Rutayisire, Rachel Sklar, Jamie Bartram, Barbara Evans.

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
