## [Decision Letter · Decision Letter 0]

10 May 2024

PONE-D-24-04267Cross-subsidies are a viable option to fund formal pit latrine emptying services: evidence from Kigali, RwandaPLOS ONE

Dear Dr. Wilcox,

Thank you for submitting your manuscript to PLOS ONE. After careful consideration, we feel that it has merit but does not fully meet PLOS ONE’s publication criteria as it currently stands. Therefore, we invite you to submit a revised version of the manuscript that addresses the points raised during the review process.

Please consider all comments 

Please submit your revised manuscript by Jun 24 2024 11:59PM If you will need more time than this to complete your revisions, please reply to this message or contact the journal office at plosone@plos.org. Please include the following items when submitting your revised manuscript:A rebuttal letter that responds to each point raised by the academic editor and reviewer(s). You should upload this letter as a separate file labeled 'Response to Reviewers'.A marked-up copy of your manuscript that highlights changes made to the original version. You should upload this as a separate file labeled 'Revised Manuscript with Track Changes'.An unmarked version of your revised paper without tracked changes. You should upload this as a separate file labeled 'Manuscript'.

We look forward to receiving your revised manuscript.

Kind regards,

Ahmed Mancy Mosa, Ph.D.

Academic Editor

PLOS ONE

Journal Requirements:

Reviewers' comments:

Reviewer's Responses to Questions

**Comments to the Author**

1. Is the manuscript technically sound, and do the data support the conclusions?

Reviewer #1: Yes

Reviewer #2: Yes

2. Has the statistical analysis been performed appropriately and rigorously? 

Reviewer #1: N/A

Reviewer #2: I Don't Know

3. Have the authors made all data underlying the findings in their manuscript fully available?

Reviewer #1: Yes

Reviewer #2: Yes

4. Is the manuscript presented in an intelligible fashion and written in standard English?

Reviewer #1: Yes

Reviewer #2: Yes

5. Review Comments to the Author

Reviewer #1: Comments for the Paper PONE-D-24-04267 “Cross-subsidies are a viable option to fund formal pit latrine emptying services: evidence from Kigali, Rwanda”

Comments

This study focuses on the role of cross-subsidies as a viable option to fund formal pit latrine emptying services. I think the paper fits well the scope of the journal and addresses an important subject. However, a number of revisions are required before the paper can be considered for publication. There are some weak points that have to be strengthened. Below please find more specific comments:

*Abstract: The abstract reads well for the most part. I particular, I suggest adding a couple of sentences highlighting the contributions of this work to the state of the art and the major potential implications from the conducted research.

*The overview of the relevant literature seems to be quite short. Please double check for the most recent and relevant studies published over the last 2-3 years. It might be a good idea to allocate a separate section, which is specifically devoted to the literature review.

*Please make sure that all the adopted assumptions are supported by the relevant references. This will help justifying the adoption of these assumptions.

*Research Method: Please provide a more thorough discussion in order to better justify the adopted of the selected research method. More supporting references might be helpful to the future readers.

*Future research: Please provide more details regarding the limitations of this study. Then, it would be good to create a more detailed discussion regarding the future research needs. I suggest listing a set of bullet points for the future research needs. This would look more appealing to the future readers.

Reviewer #2: 1. It’s recommended to include some data results in the abstract.

2. The context of your study should be placed in the Introduction section. In the Methods and Materials section, only the research methodology, modelling, and data analysis methods are detailed.

3. Have you considered the transport path optimization to reduce the cost?

4. Differences in fuel consumption exist between different models of trucks, even for the same type of truck mentioned in the manuscript, such as large exhauster trucks. Should you discuss this in the manuscript?

5. In Figure 3, it looks like the "Chart Area" should not appear in the figure, please re-upload it.

6. There are many assumptions in the model. Is it still applicable and compatible in real cases?

6. PLOS authors have the option to publish the peer review history of their article (what does this mean?). If published, this will include your full peer review and any attached files.

Reviewer #1: No

Reviewer #2: No

---

## [Author Response · Author response to Decision Letter 0]

25 Jun 2024

University of Leeds, 24 June 2024

Dear Dr Ahmed Mancy Mosa,

Thank you for identifying reviewers and coordinating their responses to review this manuscript. Please see below my responses to the points raised by yourself and the two reviewers.

I also included the paper in my PhD thesis and the examiners made two specific comments which I have addressed in the manuscript and included at the end of this letter.

Sincerely, Jonathan Wilcox

Editor comments

1. When submitting your revision, we need you to address these additional requirements. Please ensure that your manuscript meets PLOS ONE's style requirements, including those for file naming.

Confirmed.

2. Please include a complete copy of PLOS’ questionnaire on inclusivity in global research in your revised manuscript.

I have completed the questionnaire (please see updated submission) and include a sentence in the ‘Methods’ section detailing that permission was provided by the service provider which collected the data for it to be used in the study.

Reviewer 1 comments

3. Abstract: The abstract reads well for the most part. I particular, I suggest adding a couple of sentences highlighting the contributions of this work to the state of the art and the major potential implications from the conducted research.

I have added a sentence to the abstract highlighting the specific gap to imply the contribution that is made i.e. that there are no academic studies assessing this funding mechanism.

I have added a sentence highlighting the major implication of the findings (that cross subsidies are a viable funding mechanism).

4. The overview of the relevant literature seems to be quite short. Please double check for the most recent and relevant studies published over the last 2-3 years. It might be a good idea to allocate a separate section, which is specifically devoted to the literature review.

In the second paragraph of the introduction I summarise the relevant literature and identify a research gap based on recent academic and grey literature. This includes references from every year since 2016. 

PLOS guidance for introductions is to include a “brief review of the key literature” which I believe the second paragraph satisfies. 

5. Please make sure that all the adopted assumptions are supported by the relevant references. This will help justifying the adoption of these assumptions.

The modelling assumptions are already supported by eight references where values are taken from literature. Are there specific concerns about any of the assumptions made?

6. Research Method: Please provide a more thorough discussion in order to better justify the adopted of the selected research method. More supporting references might be helpful to the future readers.

I have added an additional paragraph to the beginning of the ‘Modelling assumptions’ section to introduction the overall modelling method.

7. Future research: Please provide more details regarding the limitations of this study. Then, it would be good to create a more detailed discussion regarding the future research needs. I suggest listing a set of bullet points for the future research needs. This would look more appealing to the future readers.

The ‘Limitations’ section is ~200 words long. What further details are required? I have added an additional sentence regarding generalising results to other contexts.

The future research needs are included in the conclusion.

Reviewer 2 comments

8. It’s recommended to include some data results in the abstract.

I added a financial value for the cross-subsidy required. I chose to exclude other numeral results (e.g. profit margins, revenue or costs) because they are context specific. 

9. The context of your study should be placed in the Introduction section. In the Methods and Materials section, only the research methodology, modelling, and data analysis methods are detailed.

I included details about the study context in the method section as this was the format adopted by a previous paper in PLOS (Peletz et al, 2020, When pits fill up: Supply and demand for safe pit-emptying services in Kisumu, Kenya)

10. Have you considered the transport path optimization to reduce the cost?

No. This is a good point. I have added a sentence to the limitations section of the discussion and included two references. 

11. Differences in fuel consumption exist between different models of trucks, even for the same type of truck mentioned in the manuscript, such as large exhauster trucks. Should you discuss this in the manuscript?

The difference in fuel consumption between vehicles is highlighted in the ‘Methods and materials’ section (lines 107-108), in the ‘Results’ section (line 277), and in the ‘Discussion’ section (line 448).

12. In Figure 3, it looks like the "Chart Area" should not appear in the figure, please re-upload it.

This has been corrected.

13. There are many assumptions in the model. Is it still applicable and compatible in real cases?

Are there any specific concerns with any of the assumptions?

Whilst there are several assumptions, this is also the only study I am aware which uses empirical data from a service provider, and data that is based on operational records, rather than self-reported estimates from service providers. Perhaps this was not made sufficiently prominent in the introduction?

I have added an additional sentence to abstract stating the value of the data, and the contribution this makes. 

PhD examiner comments

14. Clarify what a social enterprise is and whether it is allowed to generate profit. Elaborate then how this would be representative of for-profit companies providing the service.

I have added a sentence (lines 94-96) explaining the term social enterprise.

15. Correct the reference to costs not being considered for emptying requests that were not converted to completed jobs

Lines 144-145 were added to describe how the costs for unconverted jobs are included in analysis.

---

## [Decision Letter · Decision Letter 1]

8 Jul 2024

Cross-subsidies are a viable option to fund formal pit latrine emptying services: evidence from Kigali, Rwanda

PONE-D-24-04267R1

Dear Dr. Wilcox,

We’re pleased to inform you that your manuscript has been judged scientifically suitable for publication and will be formally accepted for publication once it meets all outstanding technical requirements.

Kind regards,

Ahmed Mancy Mosa, Ph.D.

Academic Editor

PLOS ONE

Additional Editor Comments (optional):

Reviewers' comments:

Reviewer's Responses to Questions

**Comments to the Author**

1. If the authors have adequately addressed your comments raised in a previous round of review and you feel that this manuscript is now acceptable for publication, you may indicate that here to bypass the “Comments to the Author” section, enter your conflict of interest statement in the “Confidential to Editor” section, and submit your "Accept" recommendation.

Reviewer #1: All comments have been addressed

Reviewer #2: All comments have been addressed

2. Is the manuscript technically sound, and do the data support the conclusions?

Reviewer #1: Yes

Reviewer #2: Yes

3. Has the statistical analysis been performed appropriately and rigorously? 

Reviewer #1: Yes

Reviewer #2: N/A

4. Have the authors made all data underlying the findings in their manuscript fully available?

Reviewer #1: Yes

Reviewer #2: Yes

5. Is the manuscript presented in an intelligible fashion and written in standard English?

Reviewer #1: Yes

Reviewer #2: Yes

6. Review Comments to the Author

Reviewer #1: My comments have been adequately addressed.

My comments have been adequately addressed.

My comments have been adequately addressed.

Reviewer #2: (No Response)

7. PLOS authors have the option to publish the peer review history of their article (what does this mean?). If published, this will include your full peer review and any attached files.

Reviewer #1: No

Reviewer #2: No

---

## [Editor Report · Acceptance letter]

12 Jul 2024

PONE-D-24-04267R1 

PLOS ONE

Dear Dr. Wilcox, 

I'm pleased to inform you that your manuscript has been deemed suitable for publication in PLOS ONE. Congratulations! Your manuscript is now being handed over to our production team.

Kind regards, 

on behalf of

Dr. Ahmed Mancy Mosa 

Academic Editor

PLOS ONE